# Key Learnings from the Development and Early Use of Global Guidance on the Integration of COVID-19 Vaccination into Broader Health Systems

**DOI:** 10.3390/vaccines12020196

**Published:** 2024-02-14

**Authors:** Ibrahim Dadari, Alba Vilajeliu, Viorica Berdaga, Shalini Rozario, Phoebe Meyer, Laura Nic Lochlainn, Dirk Horemans, Nuria Toro, Gloria Lihemo, Sanjay Bhardwaj, Peter Cowley, Diana Chang Blanc, Florence Conteh-Nordman, Imran Mirza, Shahira Malm, Ida Marie Ameda, Ann Lindstrand

**Affiliations:** 1Immunization Section, United Nations Children’s Fund (UNICEF) HQ, New York, NY 10017, USA; 2Essential Programme on Immunization Unit, Immunization Vaccines & Biologicals Department, World Health Organization (WHO), 1202 Geneva, Switzerland; avilajeliu@who.int (A.V.);; 3Integrated Health Services Department, World Health Organization (WHO), 1202 Geneva, Switzerland; 4Health Systems Governance and Financing, World Health Organization (WHO), 1202 Geneva, Switzerland; 5UNICEF Eastern and Southern Africa Regional Office, Nairobi P.O. Box 44145, Kenya

**Keywords:** COVID-19 vaccine, immunization programs, global guidance, integration, primary health care, integrated health services, health systems

## Abstract

More than 13.5 billion COVID-19 vaccine doses were delivered between 2021 and 2023 through a mix of delivery platforms, with mass vaccination campaigns being the main approach. In 2022, with the continued circulation of SARS-CoV2 and the need for periodic boosters being most likely, countries were required to plan for more sustainable approaches to provide COVID-19 vaccinations. In this context of uncertainty, a global tool for integrating COVID-19 vaccines into immunization programs and as part of broader health systems was published jointly by the WHO and UNICEF to respond to country needs. This paper summarizes the approach to, and lessons learned during, the development of a global guidance document and describes some examples of its early use in low- and middle-income countries (LMICs). The guidance leveraged existing health system frameworks, proposed four steps for planning and implementing the COVID-19 vaccination integration journey, and identified investment areas. The development process maximized robust global stakeholder and country engagement, and the timeframe was aligned with donor funding windows to support countries with the integration of COVID-19 vaccination. The rapid dissemination of the guidance document allowed countries to ascertain their readiness for integrating COVID-19 vaccination and inform the development of national plans and funding applications. While progress has been made in specific areas (e.g., optimizing cold chain and logistics leveraging COVID-19 vaccination), in the context of decreasing demand for COVID-19 vaccines, reaching adult COVID-19 vaccine high-priority-use groups and engaging and coordinating with other health programs (beyond immunization) remain challenges, particularly in LMICs. We share the learning that despite the uncertainties of a pandemic, guidance documents can be developed and used within a short timeframe. Working in partnership with stakeholders within and beyond immunization towards a common objective is powerful and can allow progress to be made in terms of integrating health services and better preparing for future pandemics.

## 1. Introduction and Rationale for Developing the Guidance

The COVID-19 pandemic, which was declared a public health emergency of international concern (PHEIC) in January 2020, has had an unprecedented global effect, with over 700 million cases and about 7 million deaths recorded worldwide as of December 2023 [1,2]. The pandemic and its response, including the range of public health and social measures implemented, had immense socio-economic impacts [3]. Primary health care (PHC) services, including immunization, were forced to adapt to the additional burden of pandemic response, with service disruptions ensuing [4,5,6,7,8,9,10,11].

As part of the COVID-19 pandemic response efforts, over 13.5 billion doses of COVID-19 vaccines have been administered globally within the shortest time ever for the introduction of a new vaccine [1,12,13]. The COVID-19 pandemic required a departure from ‘business-as-usual’ vaccine delivery approaches to rapidly reduce morbidity and mortality, protect health systems, and restore socio-economic activities as quickly as possible [14]. Mass vaccination campaigns were used as a main delivery approach to reach target populations quickly and widely. This approach put a strain on essential immunization and other programs [5,6,7,8,10,12,15]. Health workers and resources were diverted away from the provision of essential health services. Furthermore, it is estimated that mass vaccination campaigns cost three times as much as the delivery of the COVID-19 vaccine in routine services (Based on estimates of the COVID-19 Vaccine Delivery Introduction and Deployment Costing Tool (CVIC tool)), and some countries had to rely on donor funding, which hampered sustainability. In some instances, the rapid response required parallel pandemic response coordination structures outside immunization programs. Consequently, there was a need for guidance on the planning and implementation of a more sustainable harmonized approach to delivering COVID-19 vaccines, as part of broader health systems, such as PHC.

By 2022, some countries had already taken steps towards integrating COVID-19 vaccination and exploring new entry points for COVID-19 vaccination to identify and reach high-priority populations, mainly adults [16,17]. By then, the trajectory of the COVID-19 pandemic remained uncertain, and the WHO laid out possible scenarios for how it could evolve. With the most likely scenario being the continued circulation of the SARS-CoV-2 virus and the need for periodic booster doses (particularly for those groups at higher risk of hospitalization and death) [18], the WHO and UNICEF worked together to develop a guidance document outlining considerations for the integration of COVID-19 vaccination into immunization programs and PHC [16]. This was also critical to guide government and donor investments to mainstream COVID-19 vaccination. Finally, integrating COVID-19 vaccination within other health service delivery platforms and programs increases the opportunity for a more people-centered approach by delivering packages of health services that better respond to people’s needs across their life course, in alignment with the goals of the Immunization Agenda 2030 (IA2030) and PHC agenda [19].

This paper summarizes the approach to, and lessons learned during, the development of a global guidance document in an acute phase of the COVID-19 pandemic and describes some examples of its early use in low- and middle-income economy countries (LMICs). These lessons are expected to inform the development and promote the use of global guidance for future pandemics that require vaccination, and to inspire guidance for the integration of vertical programs into national health systems.

## 2. Design: Approach to Developing the Guidance

The development of the global COVID-19 integration guidance aligned with the steps of both the WHO-INTEGRATE and the GRADE Evidence to Decision (EtD) frameworks [20,21,22]. The development process involved an iterative process of the three cardinal steps mentioned in the GRADE EtD, including (1) problem definition; (2) evidence-informed assessments; and (3) drawing conclusions, as indicated in Figure 1. Field experiences emerging from countries indicated that some opportunistic integration of COVID-19 vaccination was occurring. The document leveraged more than 40 countries’ experiences of integrating COVID-19 vaccination, structured and analyzed by the different health system building blocks [17,23,24], which helped define the problems that needed to be addressed. EtDs principles were used in summarizing and deciding on evidence to include in the guidance, including the quality of evidence, feasibility, equity, and ethical considerations.

A joint task force with membership across UNICEF and the WHO was constituted in January 2022 to lead efforts to develop and disseminate the guidance. The methodology deployed by the task force can be summarized into four main work streams conducted in parallel and is described in detail in Annex 1 of the guidance document [16]. These were as follows:

### 2.1. Workstream 1—Publications Scoping

This involved a review of the WHO and UNICEF documents described as guidance, manuals, or tools that refer to the integration of health interventions or other health programs, or vaccination relevant to immunization, COVID-19, PHC, and integration, published after 2015. This scoping review was conducted to ensure consistency with previous publications related to integration, bringing together relevant guidance into one document that cohesively provides considerations for COVID-19 vaccination integration. These documents informed the definition and principles of integration, the scope of the document, benefits, risk analysis, and operationalization.

### 2.2. Workstream 2—Rapid Survey and Interviews

To better understand the status of COVID-19 vaccine integration across countries, including its perceived opportunities and risks, the WHO and UNICEF conducted a concurrent survey targeting country and regional office colleagues. A semistructured questionnaire was developed that comprised both Likert-like and open-ended questions developed and uploaded to Survey Monkey, and responses were received from the WHO (five regional offices and 41 country offices) and UNICEF (six regional offices and 34 country offices). The survey findings shed light on the benefits and risks associated with integration.

### 2.3. Workstream 3—Stakeholder Consultations

Consultations were conducted at the WHO and UNICEF offices (at headquarters, regional, and country level), Gavi, and other global health stakeholders. They included immunization, social behavior change, and health system experts to enable a better understanding of the benefits and risks of the integration of COVID-19 vaccination, and how integration had already been achieved in different settings. Additional country examples of the integration of COVID-19 vaccination for each of the health system building blocks, and demand promotion, including community engagement, were compiled. The findings from these consultations contributed to the sections on country experiences, how to operationalize integration, and generating further evidence.

### 2.4. Workstream 4—Draft Development

Based on the publications review, rapid survey, and consultations, an initial version was drafted and circulated to the WHO, UNICEF, Gavi, and other relevant stakeholders for input and comments. The working draft, including a readiness assessment checklist, was uploaded to a shared folder, and the task team reviewed and strengthened the guidance based on the suggestions received. A more advanced version of the guidance was discussed with immunization and health system experts who were members of the IA2030—Strategic Priority 1: PHC working group and Strategic Priority 4: Life Course and Integration working group. The comments and inputs received were consolidated and discussed by the task team, and decisions were made to include or expunge suggestions. Workstreams 3 and 4 were iterative and built on each other. A professional writer carried out a detailed review and copy-editing. Most of the costs of developing this guidance were staff time across agencies, with little or no added financial implications.

## 3. Implementation: Evidence from Early Use in LMICs

The first version of the guidance was published in July 2022, and it proposed four steps towards integrating COVID-19 vaccination as described in Box 1:
Box 1**High-Level Steps for a COVID-19 Vaccination Integration Process** **Steps 1—Initiating Integration**: Align or repurpose existing working groups and conduct multisectoral consultations and situational analysis (proposes the use of the readiness assessment checklist included in the guidance). **Step 2—Planning and Preparatory Phase:** Develop a country-led COVID-19 vaccination integration plan, including defining a national vaccination policy and service delivery strategies and interventions, and identify the key actions and investments required.**Step 3—Implementation and Monitoring:** Define indicators and targets to monitor progress and oversee the implementation of the integration.**Step 4—Postintegration Follow-up Actions:** Define the learning research agenda and explore the conduct of a postintegration evaluation.

This version was broadly disseminated and presented at several webinars, meetings, workshops, and other relevant forums targeting global, regional, national, and subnational stakeholders. Documentations of its early use are summarized below.

### 3.1. Use of the Guidance to Support Country Readiness Assessment of COVID-19 Vaccine Integration

Data on the countries’ self-reported readiness for integration were presented during the ‘COVID-19 vaccine stock take meeting’ in May 2023, facilitated by the COVID-19 Vaccine Delivery Partnership (CoVDP). The CoVDP was a partnership between the WHO, UNICEF, and Gavi to support countries in their COVID-19 vaccine delivery efforts, with a focus on the 34 countries with the lowest coverage as of January 2022. The event saw the participation of more than 30 countries, most of which used the guidance checklist (in advance of the meeting) [25], which facilitated the identification of current and anticipated enablers and challenges, and a way forward for the integration of COVID-19 vaccination into PHC. The readiness checklist allows for the self-assessment of between four and nine specific action items under eight blocks (leadership, health financing, demand and community engagement, service delivery, health workforce, health information system, access to essential medicines, and monitoring and evaluation). An overall score is proportioned based on the number of actions “completed”, “in progress”, or “not started”. A summary of the country readiness data as presented in the meeting is shown in Figure 2 below.

There was a wide variation in the self-reported readiness of countries to integrate COVID-19 vaccines, and their status may have changed since May 2023. While countries such as Chad, Djibouti, Gabon, Mali, and Niger reported relatively low readiness for integration or the initiation of steps towards integration, others, including Nigeria, Senegal, and Uganda, reported higher readiness.

The enablers identified by countries that reported being mostly ready included the presence of a COVID-19 vaccination integration plan and budgetary allocations. Others mentioned integration as necessary for sustaining their PHC services and leveraging the use of existing programs to boost COVID-19 vaccination. The challenges reported included an absence of a holistic national strategy for COVID-19 vaccine integration and competing priorities, including other disease outbreak responses and natural disasters. Some countries requested additional technical assistance to support the prioritization and development or rollout of their integration plans in alignment with the global guidance.

### 3.2. Inclusion of Integration of COVID-19 Vaccination in Country Funding Proposals

Beginning in 2021, Gavi provided operational funding to support countries in their COVID-19 vaccination efforts through the COVAX initiative. With the evolution of the pandemic and the emerging funding needs of countries, the third window of COVID-19 vaccine delivery support (CDS3) allowed countries to seek support for three objectives: (1) acceleration of vaccination of high- and highest-risk populations (as defined by the WHO Strategic Advisory Group of Experts on Immunization (SAGE)); (2) rapid delivery scale-up to reach country targets for adult vaccination; and (3) integration of COVID-19 vaccination into routine immunization (RI) to achieve sustainable benefits. The first and second windows of support (CDS1 and CDS2) did not include the integration of COVID-19 vaccination in their scope. The CDS3 timeframe for 91 eligible LMICs to submit their proposals to Gavi was between September and November 2022. We hypothesize that the existence of programmatic guidance increased the appetite of countries to plan for the integration of COVID-19 vaccination into broader health systems, aided by the several discussions and engagements which occurred following the dissemination of the guidance. Of the 52 CDS3 applications from Gavi-eligible countries, an estimated 53% were made to support the integration of COVID-19 vaccination into routine immunization and PHC, 22% related to the acceleration of the vaccination of high-risk group populations, and 30% related to rapid scale-up to reach national COVID-19 vaccine coverage targets. The prioritized activities included strengthening COVID-19 vaccine delivery strategies (including primary care services and outreach/home visits, and engagement with the private sector, e.g., pharmacies); updating operational guidelines to standardize the integration of COVID-19 vaccination into RI; developing integrated microplanning to promote the use of primary care services at the community level to sustain immunization coverage, including data collection of COVID-19 vaccination into RI information systems; upgrading the capacity of cold chain and logistics systems for the integrated management of RI and COVID-19 vaccination (including bundled vaccine and supply transportation); and updating social and behavioral change communication strategies (e.g., faith networks, community, and local influencers, strengthened social listening mechanisms to improve understanding of community concerns regarding integrated services).

A deep-dive into the postimplementation data from 30 non-Gavi-eligible countries that submitted proposals for CDS3 (referred to as AMC30) showed that close to half of the resources requested (47%) were to support COVID-19 integration efforts (Figure 3). Advanced Market Commitment (AMC) countries could apply to COVAX CDS if they were eligible for Gavi support (that is, if their most recent gross national income [GNI] per capita was less than or equal to USD 1730). AMC countries (non-Gavi-eligible) with a GNI above USD 1730 and below USD 4000 (and an additional set of IDA-eligible countries) could apply for and receive CDS through the UNICEF-managed CDS envelope—these are referred to as ‘AMC30’ countries) Despite the high proportion of activities related to integration, there was a lot of variation in how countries defined integration activities. Activities were largely planned to support the strengthening of supply chain management and supplies, planning and coordination, and human resource surge and deployment.

When asked to report on whether there has been a decision to integrate COVID-19 vaccination into routine immunization or primary care systems, fifteen (50%) of the countries responded that there was a plan or decision to integrate COVID-19 vaccination, and a few had already included COVID-19 vaccines into their national immunization programs (e.g., Bolivia, the Republic of Moldova, and Sri Lanka) [26]. Countries continue to face operational challenges, including balancing in-country procedures, and the complexity of coordination, logistics, and rapid scale-up within the limited amount of time available to implement emergency resources. Acknowledging that the nature of CDS3-funded activities, particularly regarding COVID-19 vaccine integration, requires longer timelines for implementation, alignment with competing priorities and processes within countries, extensive stakeholder mapping, engagement, and costing support, the Gavi Board approved a noncost extension of CDS3 for eligible countries for 2024 and 2025 [27].

## 4. Discussion

The continuous evolution of SARS-CoV-2, with the frequent emergence of new variants, has often created a situation requiring the rapid development or continuous updating of guidance and tools in a context of high uncertainty. Like other guidance documents developed during the pandemic, the COVID-19 vaccination integration global guidance was developed without the privilege of face-to-face interaction. However [28], the overall guidance development and application process aligned well with the available EtD frameworks [20,21,22]. Some of the integration efforts that happened ahead of the dissemination of the guidance were more opportunistic than strategic, with countries reporting facing challenges in operationalizing the integration of COVID-19 vaccination. Informed by these countries’ experiences and leveraging existing health system frameworks, the guidance identified four concrete steps for countries to initiate and plan for integration, as listed earlier. These steps were consolidated following robust engagement and feedback and simplified to make the process more practicable and agile for countries. The concept of initiating integration and preparatory work that includes a readiness assessment provides a robust foundation for planning and implementing COVID-19 vaccine integration. As shown by the early use evidence, several LMICs were able to adapt the readiness assessment and report to meet their needs and confidently report their status. In addition, mapping the necessary short-term (6–12 months) investments using the health system building blocks and aligning this with the timeframes of funding opportunities enabled the prioritization of COVID-19 vaccination integration in country plans and funding proposals. This could suggest that the guidance spoke to the country’s needs. It is anticipated that these countries should also be able to develop and implement their respective post integration follow-up actions.

Co-development in partnership with stakeholders at the national, regional, and global levels beyond immunization was key to jointly defining a new concept: integrating COVID-19 vaccination into immunization programs and other PHC services. As countries continue to integrate COVID-19 vaccines, feedback to enable further simplification of and improvement in the definition is welcome. The experience gained in the development of the Integration guidance further demonstrated the resilience and commitment of global health partnerships, which remain focused on addressing the overall public health needs of the prevention and control of the COVID-19 pandemic in the face of rapidly changing circumstances. Thus, waiting for a ‘suitable’ time was not—and will never be—feasible.

While the guidance was developed as a ‘living document’ in 2022, it remains relevant for the present and the future [15,29,30]. The COVID-19 vaccination effort did not end with the standing down of the PHEIC in May 2023. SARS-CoV-2 is still circulating, with new variants emerging, and certain groups continue to be at greater risk of severe disease and death. In 2023, the WHO SAGE recommended a simplified single-dose regime for primary vaccination for most COVID-19 vaccines and focusing on increasing the vaccination coverage of older adults, adults with comorbidities and severe obesity, people with immunocompromising conditions, pregnant people, and frontline health workers [31]. In alignment with this, the WHO’s post-PHEIC recommendations called for the enhancement of elements of future preparedness and the integration of COVID-19 vaccination into life course vaccination (mainly targeting adult populations), which includes the sustainability and resilience of service delivery platforms [32]. The most recent SAGE recommendation is for a shift towards a simplified single-dose regime for primary vaccination for most COVID-19 vaccines. In consideration of these recommendations and the need for the long-term planning of COVID-19 vaccination, the global guidance has been used to develop regional strategies. The WHO Regional Office for Europe recently published a shorter version of the global guidance on developing a national COVID-19 vaccination policy and integrating COVID-19 vaccination into national immunization programs and broader healthcare delivery mechanisms [33], the Pan American Health Organization/WHO Regional Office for the Americas promoted the co-administration of COVID-19 vaccine with other vaccines (particularly influenza) and health interventions as part of their routine primary care services, and the WHO African Region leveraged the approach proposed in the global guidance to inform its COVID-19 Strategic Preparedness and Response Plan for 2023–2025 [34]. The integration guidance also speaks to the need to monitor and assess the progress on integration. While effectively engaging other health programs and ministries beyond immunization continues to be difficult and remains a work in progress, the lessons learnt are expected to inform the introduction of future vaccines that target adults; prepare for future pandemics; and develop approaches and guidance for the integration of other vertical PHC services [35]. Despite the fact that countries may have previously faced challenges with funding to drive integration, due to much of the earlier pandemic response funding being ringfenced, the 2024 and 2025 Gavi COVID-19 vaccination program targeting 91 LMICs offers opportunities for improved integration [27].

However, the unclear long-term COVID-19 vaccine policy recommendations, the decrease in the risk perception of COVID-19 by the population (including high-priority-use groups) and misinformation, the lack of secured long-term funding, and the emergence of competing country priorities remain existential threats to this agenda. COVID-19 vaccines continue to be the most effective tool to prevent serious illness, hospitalization, and death from COVID-19, and leveraging the COVID-19 legacy affords a unique opportunity to contribute to resilient programs, Immunization Agenda 2030 goals, and the PHC agenda.

Some of the limitations of this manuscript include that the guidance document is still in its early implementation phase, having been published in July 2022, and it will take time to document lessons learnt and progress made across a diverse range of countries. The country examples are from LMICs due to the focus of global health partners in supporting improving COVID-19 vaccination coverage and integration in these countries. Secondly, some of the lessons being documented are based on the available funding and technical assistance support from global health partners, with the likelihood that the picture might be different in the event of fragile and weak health systems having limited support for the integration process. Finally, since no specific systems exist to monitor the use of the guidance, and as no formal evaluation has been conducted, our paper presents the lessons learnt based on the available data and evidence obtained so far.

## 5. Conclusions

While no formal evaluation has been conducted, the rapid process for developing the WHO–UNICEF global guidance for integrating COVID-19 vaccination as part of broader health systems in collaboration with many stakeholders beyond immunization, along with the broad dissemination of information to countries and alignment with funding opportunities, suggests that this enabled its early use by countries. COVID-19 vaccination has become an important element of health programs globally, and it is important to continue monitoring and learning how countries move towards mainstreaming COVID-19 vaccines as an element of PHC and other relevant health services across the life course and as an element for better pandemic preparedness and response in the future.

## Figures and Tables

**Figure 1 vaccines-12-00196-f001:**
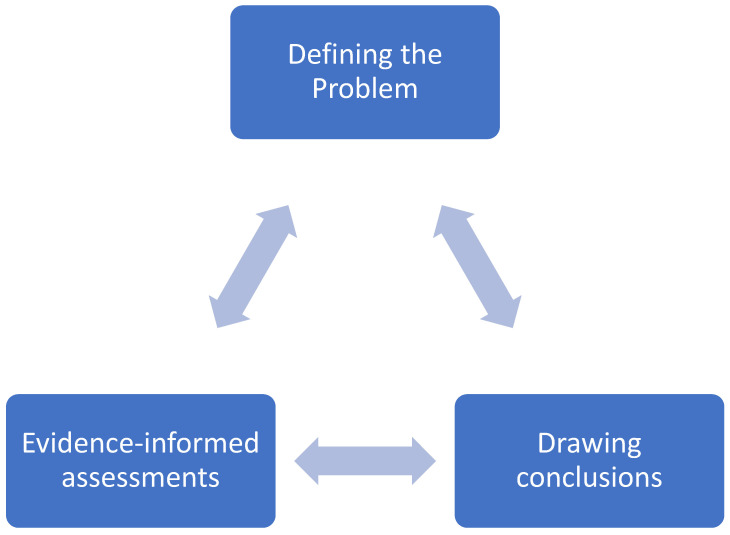
Schematic of the iterative process used in developing the global guidance for integrating COVID-19 vaccination into immunization programs and PHC.

**Figure 2 vaccines-12-00196-f002:**
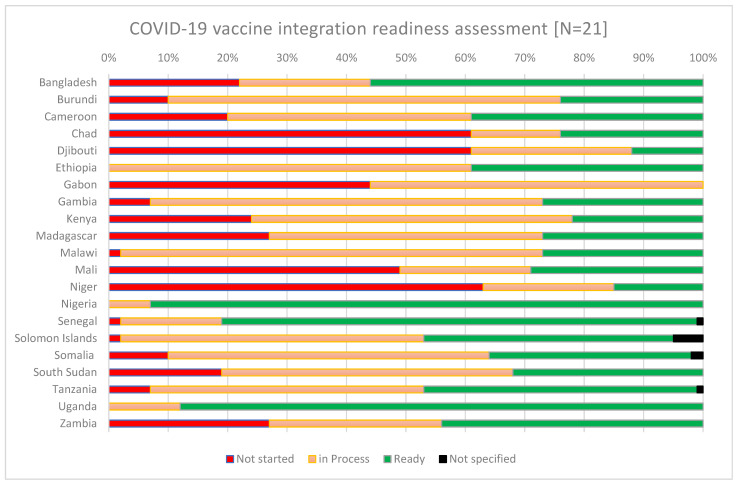
Country self-reported readiness status for COVID-19 vaccine integration into PHC and other health services [status as of May 2023].

**Figure 3 vaccines-12-00196-f003:**
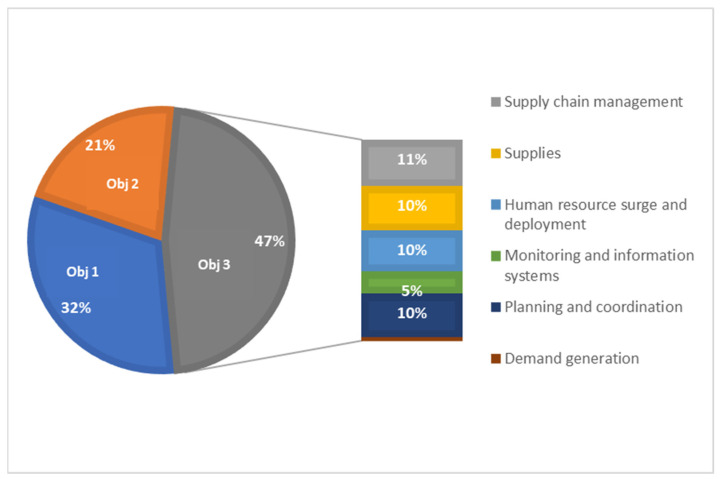
COVID-19 vaccine delivery support (CDS) needs directed towards COVID-19 integration activities.

## Data Availability

The data used in this paper have been cited and are publicly available.

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
