# Peer review of "Key Learnings from the Development and Early Use of Global Guidance on the Integration of COVID-19 Vaccination into Broader Health Systems"

_vaccines, 2024, doi:10.3390/vaccines12020196_

Round 1

Reviewer 1 Report

Comments and Suggestions for Authors

Is this report of interest for the average reader? Does the description of the development of a report like this enrich the reader's understanding of the magnitude and scope of the problem?

The statements in lines 64-66 and 328-333 are rather wishes than facts. I miss the mention of vaccine opposition in the internet era, the issue of child vaccination, and comparing affluent countries' (like Israel) response with LMIC (Like sub-Sahara countries).

I want to ask about the total cost of the development of this document, and the lives saved in LMIC.

How did the authors and other participants avoid the strong lobby of the producers of the vaccines.

Author Response

Dear Reviewer,

Many thanks for your valuable insights, please find in the attached document responses to your comments. We hope this provides the needed clarity and input. 

Reviewer 2 Report

Comments and Suggestions for Authors

The study designed a tool to guide the integration of COVID-19 vaccination into immunization programmes and primary health care. I have the following concerns and comments.

1. The tool was designed targeting multiple regions and countries. However, it seems that the tool might focus more on the countries with lowest coverage of COVID-19 vaccines. I believe high-income countries and other low- and middle-income countries have specific country needs, though they had much higher coverage ov COVID-19 vaccines. Thus, I wonder if the tool would be applicable to diverse regions and countries due to diverse contexts and health care systems, or the title might be revised to demonstrate the guidance for low- and middle-income countries.

2. The authors presented the Applications: Evidence from early use, as the tool was published in Jly 2022. However, the authors should elaborate more on how to apply the tool and then improve it in 3.2 section. I suggest some tables, figures, or processes.

3. Please add the study limitations in the manuscript.

Author Response

(The authors gave the same response as above.)

Reviewer 3 Report

Comments and Suggestions for Authors

I would like to express my gratitude to the Editor for providing me with the opportunity to review the manuscript.
Although I also believe that the topic is of interest to the scientific community, I admit that the manner in which it has been presented and discussed seemed confusing to me. I do not think it qualifies as an original article, contrary to what is stated in the manuscript; rather, it appears to be more of a comment or letter to the Editors. If this is the case, a thorough review of the presentation and paragraph division will certainly be necessary.

The title is not very clear and does not fully reflect what is being addressed. The abstract, in my opinion, should be more organized and more dedicated to the methodological aspects. While the introduction is well-written, it is extremely concise.

What perplexes me the most is that the Methods in this manuscript are entirely inconsistent. It is unclear which steps were followed; some of them are mentioned superficially, and nothing is addressed in depth. The entire methodology is deferred to Annex 1 of the cited document. This leads me to believe that it is not an original work but rather a comment or letter to the Editors.

Sections 3.1. and 3.2. are presented in an utterly confusing manner. There is no adherence to the structure of a scientific article, questions are posed without reporting the methodology, and results are alternately presented.

The discussion further confirms that it would be more appropriate to present the fully revised contents of this manuscript in the form of a letter to the Editor.

Finally, I advise paying attention to the overstatements in the conclusion: no assessment of the effectiveness of the guidelines has been conducted.
The authors merely describe and comment on what has already been done and published.

Author Response

(The authors gave the same response as above.)

Round 2

Reviewer 2 Report

Comments and Suggestions for Authors

As the type of the paper has been changed to Project Report, I suggest the authors may further revise the manuscript, such as shortening the length of text, to highlight the main points in the design, impelmentation, and improvement of the tool.

Author Response

Dear Reviewer,

Thank you for your valuable comments. We revised the paper accordingly, reducing the text to be concise while maintaining the minimum of 3500 words for a Project Report as per the Journal requirements. 

Kind regards,

Ibrahim

Reviewer 3 Report

Comments and Suggestions for Authors

I express my gratitude to the authors for addressing my comments point by point and reconsidering the manuscript's structure. I believe it is more fitting to present it to readers as a project report rather than an original article, as it does not meet the criteria for the latter. I also appreciate the effort made to avoid overstatements in the conclusions, given the absence of any real-world efficacy testing of the guidelines.

However, I still harbor entirely personal reservations regarding the utility and relevance of this work to the scientific community.

Author Response

Dear Reviewer,

Thank you for your valuable comments. We revised the paper to make it more succinct and further bring out the value of the paper.  

Kind regards,

Ibrahim
